# Myocardial Perfusion and Coronary Physiology Assessment of Microvascular Dysfunction in Patients Undergoing Transcatheter Aortic Valve Implantation—Rationale and Design

**DOI:** 10.3390/biomimetics7040230

**Published:** 2022-12-08

**Authors:** M. M. Dobrolinska, P. Gąsior, A. Błach, R. Gocoł, D. Hudziak, W. Wojakowski

**Affiliations:** 1Department of Cardiology and Structural Heart Diseases, Medical University of Silesia in Katowice, 40-635 Katowice, Poland; 2Nuclear Medicine Department, Voxel Medical Diagnostic Centre, 40-635 Katowice, Poland; 3Department of Cardiac Surgery, Medical University of Silesia, 40-635 Katowice, Poland

**Keywords:** aortic stenosis, fractional flow reserve, index of microvascular resistance, coronary flow reserve, cadmium-zinc-telluride single-photon emission tomography (CZT-SPECT)

## Abstract

The prevalence of coronary artery disease (CAD) in patients with severe aortic stenosis (AS) is 30–68%. Nevertheless, there is still not enough evidence to use invasive assessment of lesion severity, because the hemodynamic milieu of AS may impact the fractional flow reserve (FFR) and non-hyperemic indices. Therefore, the aim of the study is two-fold. First, to measure acute and long-term changes of FFR, index of microvascular resistance (IMR), and coronary flow reserve (CFR) in patients undergoing TAVI procedure. Second, to compare the diagnostic accuracy of intracoronary indices with myocardial perfusion measured by cadmium-zinc-telluride single-photon emission tomography (CZT-SPECT) and find cut-off values defining significant stenosis. We plan to enroll 40 patients eligible for TAVI with intermediate stenosis (30–70%) in the left anterior descending (LAD) coronary artery. In each patient FFR, CFR, and IMR will be measured in addition to myocardial blood flow calculated by CZT-SPECT before and either immediately after TAVI (acute cohort) or in 6 months (late cohort) after the procedure. FFR, CFR, and IMR will be matched with the results of myocardial perfusion measured by CZT-SPECT in the area of LAD. As a result, cut-off values of FFR, CFR, and IMR defining the decreased blood flow will be found.

## 1. Introduction

Aortic stenosis (AS) is the most common valvular disease, and the number of patients is set to double in the next 20 years [1]. Importantly, in 30–68% of patients diagnosed with AS, coronary artery stenosis is also present [2].

In clinical practice, the severity of coronary artery stenosis and microvasculature dysfunction can be assessed by means of fractional flow reserve (FFR), coronary flow reserve (CFR), and index of microvascular resistance (IMR). However, in patients with AS, the assessment of coronary flow is impaired due to volume and pressure overload of the left ventricle. Moreover, in patients with AS and concomitant coronary artery disease (CAD) not a single stenosis but tandem stenosis is present—first is the aortic stenosis, and second is the coronary artery stenosis. As a result, the hyperaemic flow in coronary arteries is changed which affects the measurement of coronary artery hemodynamics [3].

Transcatheter aortic valve implantation (TAVI) allows the rapid unloading of the left ventricle and can acutely affect coronary flow. As reported by Pesarini et al. and Lunardi et al., changes in FFR after transcatheter aortic valve implantation (TAVI) were insignificant, indicating that this tool can be used to guide revascularisation in AS patients [4,5]. On the other hand, the underestimation of coronary stenosis by hyperemic indices was found in most of the studies conducted so far [6,7,8,9,10]. The discrepancy between findings and lack of evidence is reflected in current guidelines, which still maintain coronary angiography as a standard to guide revascularization [11,12].

All things considered, the physiological assessment of intermediate coronary artery stenosis severity in patients with severe AS is still a real challenge. Therefore, the aim of this study is to investigate the changes in FFR, CFR, IMR, and myocardial perfusion in patients with severe AS after TAVI. As a first endpoint, we will assess acute and long-term changes in FFR, IMR, and CFR in patients undergoing the TAVI procedure. The secondary endpoint is to determine cut-off values of intracoronary indices defining significant stenosis based on myocardial perfusion measured by cadmium-zinc-telluride single-photon emission tomography (CZT-SPECT), in patients with severe AS and concomitant CAD.

## 2. Materials and Methods

### 2.1. Study Design

We are planning to enroll 40 patients, all of whom have a clinical indication to elective TAVI, as jointly evaluated by the local Heart Team, and are diagnosed with intermediate stenosis (30–70%) in the left anterior descending (LAD) artery. We will measure FFR, CFR, and IMR before, and either immediately after (Group I) or 6 months after (Group II) TAVI. At the same time, myocardial perfusion will be assessed by CZT- SPECT, to compare intravascular indices to myocardial perfusion results (Figure 1).

### 2.2. Study Objectives

The primary objective is to measure FFR, CFR, IMF, and myocardial perfusion CZT-SPECT before the TAVI procedure and either immediately after TAVI or after 6 months, to specify the acute and long-term changes in coronary pressure and flow. The secondary objective is to compare FFR, CFR, and IMR to myocardial perfusion CZT-SPECT to identify cut-off values of CFR, IMR, and FFR defining significant stenosis (Figure 2).

### 2.3. Population

Patients referred to TAVI with the present intermediate stenosis in LAD are eligible for the study. The exclusion and inclusion criteria are summarised in Table 1. As we are interested in the influence of left ventricle volume overload decrease after TAVI, only LAD will be assessed.

### 2.4. Sample Size

Our target sample size is set at 40 patients for a proof-of-concept study. No power calculations were performed since this is currently a pilot study and relevant data on our research objectives regarding the comparison between patients suffering from AS with CFR and IMR assessment are lacking. The results of this study will be used to design a larger study in the near future. Since the total number of patients for AS diagnosis in the Division of Cardiology and Structural Heart Diseases is estimated at 100 a year, we expect that our target number of patients can be included within one year.

### 2.5. Patient Recruitment

When all inclusion criteria are met, the patient will be asked by the physician to participate in the study. After signing the informed consent, each patient will undergo rest and stress CZT-SPECT and coronary angiography with measurement of FFR, CFR, and IMR. This diagnostic process will be repeated immediately after TAVI (Group I) or 6 months later (Group II).

### 2.6. SPECT Protocol

We plan to perform a 1-day protocol. Patients will be instructed not to consume chocolate and caffeine products 24 h before the test. Dynamic MPI-SPECT will be performed on a dedicated CZT-SPECT camera (D-SPECT Cardio, Spectrum Dynamics Medical). After the patient’s heart positioning within the field of view with the use of intravenous administration of an initial dose of approx. 37 MBq of ^99m^Tc-MIBI, an intravenous bolus infusion of 3.5 MBq/kg of ^99m^Tc-MIBI at a rate of 1–2 cm^3^/s using an automatic syringe pump will start for the resting dynamic scan. After a 25 min delay, an 8 min resting static perfusion scan will be performed.

In the next stage, we will inject 0.4 mg of regadenoson for the stress phase. Regadenoson bolus will be flashed with 5 mL 0.9% NaCl and 1 min after the injection, a stress phase list-mode acquisition will start with the administration of a second bolus of ^99m^Tc-MIBI (10 MBq/kg or 3× resting dose) at peak hyperaemia. Then after a 25 min delay a stress static perfusion scan will be conducted. Rest-stress dynamic acquisitions will be completed in about 75 min. List-mode data will be rebinned into 32 frames consisting of 21 × 3 s, 1 × 9 s, 1 × 15 s, 1 × 21 s, 1 × 27 s, and 7 × 30 s frames. During the rest and stress phase, monitoring of heart rate and rhythm, blood pressure, pulse oximetry, and electrocardiography (ECG) will be performed. Myocardial perfusion will be reported based on a 17-segment model of the heart [13]. For each segment, a coronary flow reserve (CFR) will be calculated with dedicated software (Corridor 4DM, INVIA, USA).

### 2.7. FFR, CFR, IMR Protocol

During invasive coronary angiography, patients with an intermediate lesion in LAD will undergo a hemodynamic assessment with FFR, CFR, and IMR. The pressure wire (Pressure Wire X, Abbott, IL, USA) will be positioned distally to the stenosis to acquire FFR, CFR, and IMR measurements. After the administration of nitroglycerine (100–200 µm), hyperemia will be induced by infusion of adenosine (140 mg/kg/min). To measure mean transit time (T_mn_), a thermodilution curve will be obtained by using 3 injections of 4 mL of room temperature saline. Hyperemic proximal aortic pressure (P_a_), distal arterial pressure (P_d_), and hyperemic T_mn_ will be measured before and during hyperemia. CFR and IMR will be calculated with the Coroflow software (Coroventis, Uppsala, Sweden). The operator will be blinded to the results of baseline measurements and to the results of CZT-SPECT scans.

### 2.8. Calculation of FFR, IMR, and CFR

#### 2.8.1. Fractional Flow Reserve

Fractional flow reserve (FFR) is calculated as a ratio of pressure measured distally to the stenosis (P_d_) and in the aorta (P_a_) [14,15].
$$\mathrm{FFR}=\frac{P_{d}}{P_{a}}$$

#### 2.8.2. Index of Microvascular Resistance

Myocardial resistance is mainly determined by microcirculation which can be measured with a coronary guidewire and expressed as an Index of Microvascular resistance (IMR). The IMR is calculated by multiplying the distal coronary pressure (P_d_) by the T_mn_ of a saline bolus during coronary hyperaemia induced by intravenous adenosine [16].
$$\mathrm{IMR}=T_{\mathrm{mn}}\times {\;P}_{d}$$

#### 2.8.3. Coronary Flow Reserve

CFR represents the vasodilator capacity of the coronary vasculature during hyperaemia. It investigates both epicardial and microvascular diseases but does not allow discrimination between them. CFR is calculated by dividing the T_mn_ of saline at rest by the T_mn_ of saline in hyperaemia induced by intravenous adenosine [3,17,18]
$$\mathrm{CFR}=\frac{T_{\mathrm{mn}}\mathrm{Rest}}{T_{\mathrm{mn}}\mathrm{Hyp}}$$

#### 2.8.4. Calculation of Myocardial Blood Flow by SPECT

The time-activity curves (TAC) for the input function will be derived automatically after correction for motion artefacts. As a first step, to calculate myocardial retention rate a previously described model will be used [19]. As a second step, to convert the retention rate to myocardial blood flow values, the Renkin–Crone flow model will be used [19]. In the end, CFR will be calculated by dividing MBF in stress by MBF in rest. As a next step, the CFR measured by CZT-SPECT will be matched with FFR, CFR, and IMR calculated in LAD.

### 2.9. Statistical Analysis

Normally distributed quantitative variables will be presented with mean, standard deviation, and 95% confidence interval. Non-normally distributed data will be presented with median and interquartile ranges. The difference between intracoronary indices values before and after TAVI will be assessed with Student *t* or Mann–Whitney *U* tests as appropriate. Sensitivity, specificity, positive predictive value (PPV), and negative predictive value (NPV) of intracoronary indices measured in LAD will be calculated on a per-patient basis. Receiver operator characteristic (ROC) curves will be constructed for FFR, CFR, and IMR to determine diagnostic cut-off values. Correlations between intracoronary indices and MBF measured by CZT-SPECT will be assessed using Pearson’s correlations. A statistically significant difference will be defined as a two-sided *P* value less than 0.05. The missing data will be labelled.

## 3. Discussion

In our study, we will investigate the immediate and long-term changes of FFR, CFR, and IMR after TAVI, and compare these results with measurements acquired before the TAVI procedure. As a next step, we aim to validate FFR, CFR, and IMR against myocardial perfusion SPECT. Therefore, this study will help to determine the influence of severe AS on intermediate coronary artery lesion severity which is still not fully understood. As a result, intravascular indices might be used to guide decision-making on revascularisation in patients with severe AS and concomitant CAD.

The narrowed area of the aortic valvular orifice reduces the blood flow from the left ventricle and causes a pressure drop in the aorta. As a result, the pressure load in the left ventricle and within the left ventricular wall is increased and driving forces of coronary flow are decreased. The increased intraventricular pressure creates a force that reduces the blood flow from the subepicardium to the subendocardium [20]. Both reduced coronary flow during systole and decreased subendocardial perfusion cause preferential epicardial blood flow and ‘blood flow mismatch’ which results in myocardial ischaemia [21,22]. Moreover, due to the fact that FFR, CFR, and IMR are measured during the entire cardiac cycle, each of them is influenced by prolonged systole in severe AS. Therefore, the physiological assessment of coronary artery stenosis in this group of patients remains challenging.

In clinical practice, FFR is used as a gold standard to determine significant coronary artery stenosis in intermediate lesions [23,24]. In patients with severe AS, the accuracy of FFR has not been confirmed, therefore guidelines still support ICA as a gatekeeper for revascularisation in this group of patients [12]. Importantly, in patients with AS the correlation between angiographic and hemodynamic significance of coronary artery stenosis is mild and the specificity and positive predictive value of ICA in the detection of significant stenosis is low, especially in LAD [25,26].

It was previously found that FFR is underestimating coronary stenosis severity in patients before TAVI because it decreases within the same coronary lesion after the TAVI procedure [6]. Some studies reported that the change in FFR after TAVI is minor. However, as underlined by Vendrik and colleagues, these minor discrepancies are the main challenge in borderline FFR results. At the same time, these borderline lesions are also the most challenging in ICA assessment. Therefore, the main problem remains unsolved. Currently, several ongoing trials will compare FFR-guided revascularization to angiography-guided revascularization in patients with severe-AS (FAITAVI (Functional Assessment in TAVI); (NOTION-3)) which hopefully will shed light on further management of patients with severe AS and concomitant CAD. Unfortunately, none of these trial implements offer invasive indices or non-invasive imaging methods which may improve the assessment of coronary artery stenosis.

In patients with severe AS, an increased resting flow enables adequate perfusion at rest, but not during stress. That was reflected in previous studies which showed that resting blood flow velocity in patients with AS is almost 18% higher and hyperaemic flow is 34% lower as compared to the control group [27,28]. Consequently, the increased baseline flow and reduced hyperaemic flow result in CFR reduction. As far as IMR is concerned, the decrease in myocardial resistance at rest has been demonstrated in patients with AS [29]. However, there was no difference between the control group and patients with AS in hyperaemic IMR values [30]. As explained by Lumley and colleagues, ischemia in patients with severe AS was not caused by microvascular dysfunction, but by abnormal coronary coupling which affects both FFR and CFR, but not IMR [30]. All things considered, the effect of AS on coronary flow and microvasculature still needs to be clarified.

In previous studies, FFR was validated against myocardial perfusion SPECT, revealing a cut-off value higher than in patients without severe AS [31]. Importantly, so far a semiquantitative SPECT was used which does not allow for myocardial blood flow quantification. In our study, CZT -SPECT will be applied which provides a quantitative analysis of myocardial blood flow and myocardial flow reserve (MFR) which provides a more accurate assessment of multivessel and microvascular disease [32,33].

Notwithstanding the clinical value of previous studies, patients with both stenotic and non-stenotic coronary arteries were included, indices were measured in different coronary arteries and directly after the implantation of the aortic valve, and there was no comparison of all three indices with non-invasive imaging. It goes without saying that FFR, CFR, and IMR together provide a broad overview of myocardial hemodynamics which may help to understand the influence of AS on the assessment of CAD. Therefore, we aim to analyse FFR, IMR, and CFR in the left anterior descending coronary artery, immediately after TAVI and in 6 months, and compare these results to CZT-SPECT analysis. If the combination of CFR, IMF, and FFR will be accurate for the coronary assessment in patients suffering from AS, we may imply these methods in daily practice to determine patients who may benefit from revascularisation.

## 4. Limitations

This study has several limitations. First, this is a proof-of-concept study with a small sample size. Nevertheless, the number of patients enrolled enables us to derive cut-off values for intracoronary indices and to design a study with an increased sample size in the future. Second, patients will be randomized into two groups—in the first one an acute change in intracoronary indices will be analysed, and in the second one, a long-term change. The decision to randomise patients into two groups instead of analysing acute and long-term changes in each participant was to decrease the radiation exposure. Third, one may argue that magnetic resonance or positron emission tomography (PET) should be used to validate intracoronary indices. Importantly, we plan to use CZT-SPECT which enables quantitative analysis of coronary flow reserve and it was found to be a feasible tool in myocardial blood flow analysis as compared to showing a good correlation with [19] ammonia-PET [34,35].

## 5. Conclusions

Due to a great number of patients suffering from severe AS with concomitant coronary artery disease, proper diagnostic steps in the assessment of coronary artery stenosis severity still need to be established. Therefore, we aim to perform a first proof-of-concept study in which intracoronary indices will be measured before and after TAVI, and compared to quantitative myocardial perfusion CZT-SPECT. As a next step, we will define cut-off values of FFR, CFR, and IMR reflecting the decreased coronary blood flow and significant stenosis.

## Figures and Tables

**Figure 1 biomimetics-07-00230-f001:**
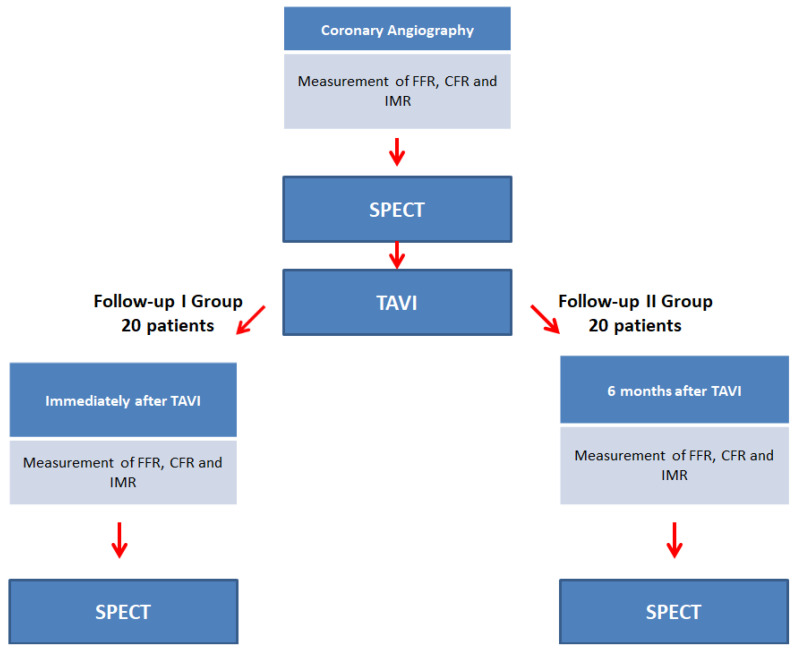
Study design and first endpoint.

**Figure 2 biomimetics-07-00230-f002:**
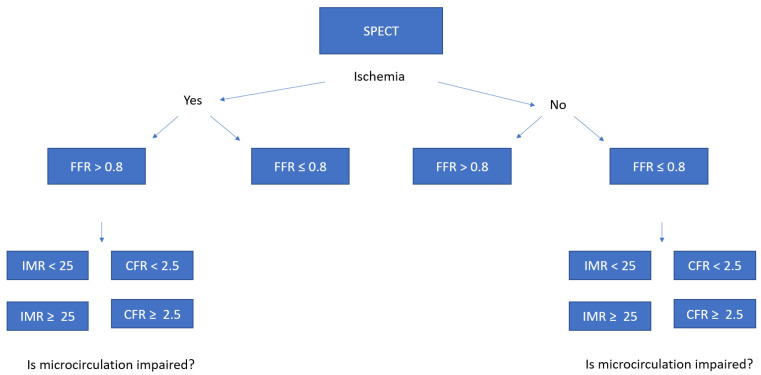
Summary of a second endpoint.

**Table 1 biomimetics-07-00230-t001:** The summary of inclusion and exclusion criteria.

Inclusion criteria
Severe aortic valve stenosis qualification for TAVI by the heart team
Presence of an intermediate (30%–70%) coronary lesion in LAD
**Exclusion criteria**
Patients who are hemodynamically unstable
LVEF < 50%
Pregnancy or lactation
Contraindications to adenosine
Chronic kidney disease (eGFR < 30 mL/min/1.73 m^2^)
Coronary artery disease which requires revascularisation
CABG in the past

## Data Availability

Not applicable.

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
