# Peer review of "Myocardial Perfusion and Coronary Physiology Assessment of Microvascular Dysfunction in Patients Undergoing Transcatheter Aortic Valve Implantation—Rationale and Design"

_biomimetics, 2022, doi:10.3390/biomimetics7040230_

Round 1

Reviewer 1 Report

Your results on FFR etc could also be influenced by an EF improvement and not necessarily from the increased flow through the aortic valve.Did you think of normalizing the calculated parameters also for EF % change? And time of modification?

Author Response

Dear Reviewer,

Thank you for this comment. The aim of this study is to investigate intracoronary indices, therefore we aim to include only patients with EF > 50%. As you noticed, in Table 1 we had a typo - EF<35% was underlined. Accordingly, we do not think that the change in ejection fraction, in patients with EF>50% at baseline, will signifficantly influence results. 

Reviewer 2 Report

"Myocardial perfusion and coronary physiology assessment of microvascular dysfunction in patients undergoing transcatheter aortic valve implantation – rationale and design" is a very interesting article from both a research and clinical point of view, as it could contribute initial results that could change current clinical practice and influence to guidelines for the management of patients before TAVI. Apart from a few minor errors, the article is very well written and understandable (line 283). I have only one methodological remark that could possibly influence clearer results. In the limitations chapter, the authors state that the reason they will not perform measurements immediately and 6 months after TAVI is to reduce possible side effects of radiation. I believe that it is necessary to carefully consider the possibility of measurements in all patients in both periods. This would certainly contribute to the assessment of the impact of TAVI on coronary artery perfusion both in the short and long term.

Author Response

Dear Reviewer,

Thank you very much for this comment, we will correct all misspellings. 

In terms of methodology, we do agree that it would be valuable to conduct all measurements twice (immdediately and 6 months after TAVI). However, due to ethical reasons, the approval from Bioethical Committee was given only for one measurement.